# One of Nature’s Puzzles Is Assembled: Analog of the Earth’s Most Complex Mineral, Ewingite, Synthesized in a Laboratory

**DOI:** 10.3390/ma15196643

**Published:** 2022-09-25

**Authors:** Olga S. Tyumentseva, Ilya V. Kornyakov, Anatoly V. Kasatkin, Jakub Plášil, Maria G. Krzhizhanovskaya, Sergey V. Krivovichev, Peter C. Burns, Vladislav V. Gurzhiy

**Affiliations:** 1Department of Crystallography, St. Petersburg State University, University Emb. 7/9, 199034 St. Petersburg, Russia; 2Laboratory of Nature-Inspired Technologies and Environmental Safety of the Arctic, Kola Science Centre, Russian Academy of Sciences, Fersmana 14, 184209 Apatity, Russia; 3Fersman Mineralogical Museum of the Russian Academy of Sciences, Leninskiy Pr. 18, 2, 119071 Apatity, Russia; 4Institute of Physics ASCR, v.v.i., Na Slovance 2, 818221 Prague, Czech Republic; 5Nanomaterials Research Centre, Kola Science Centre, Russian Academy of Sciences, Fersmana 14, 184209 Apatity, Russia; 6Department of Civil and Environmental Engineering and Earth Sciences, University of Notre Dame, Notre Dame, IN 46556, USA; 7Department of Chemistry and Biochemistry, University of Notre Dame, Notre Dame, IN 46556, USA

**Keywords:** uranyl, carbonate, ewingite, mineral, nanocluster, topology, crystal structure, structural complexity, X-ray diffraction

## Abstract

Through the combination of low-temperature hydrothermal synthesis and room-temperature evaporation, a synthetic phase similar in composition and crystal structure to the Earth’s most complex mineral, ewingite, was obtained. The crystal structures of both natural and synthetic compounds are based on supertetrahedral uranyl-carbonate nanoclusters that are arranged according to the cubic body-centered lattice principle. The structure and composition of the uranyl carbonate nanocluster were refined using the data on synthetic material. Although the stability of natural ewingite is higher (according to visual observation and experimental studies), the synthetic phase can be regarded as a primary and/or metastable reaction product which further re-crystallizes into a more stable form under environmental conditions.

## 1. Introduction

It is well known that modern preparative chemistry emerged as a new field of science from mineralogical studies about 500 years ago. One of the founding fathers of this discipline was Georgius Agricola, who worked in the city of St. Joachimsthal (modern Jáchymov, Czech Republic) in the middle of the 16th century. This region of Ore Mountains, including Joachimsthal, has long been known for its unique geological and mineralogical treasures [1,2].

Georgius Agricola brought silver mining to a novel metallurgical level [3], whereas at the end of the 19th century, Pierre and Marie Curie discovered polonium and radium from the Joachimsthal uranium ores [4], and 50 years later the same uranium ores served as the basis for the creation of the USSR’s nuclear industry. Although uranium mining ceased in 1964, its remains in the Jáchymov ore district continue to provide scientists with exceptional and diverse mineralogical samples. Many new U-bearing minerals have been described in Jáchymov recently (e.g., see [5,6,7,8,9,10,11,12]), including one of the most amazing and structurally the most complex known mineral, ewingite, Mg_8_Ca_8_[(UO_2_)_24_O_4_(OH)_12_(CO_3_)_30_](H_2_O)_138_ [13]. Ewingite was recently found in samples from the abandoned Plavno mine in the Jáchymov ore district and has an unprecedented structure based on uranyl carbonate nanoclusters. Ewingite is formed during the alteration processes of uranium ores (mainly uraninite, UO_2+x_) upon contact with carbonate-enriched groundwater, although the precise conditions and mechanisms of its formation are still unknown.

Herein, we report the synthesis and characterization of the novel synthetic compound Ca_21_(H_3_O)_6_[(UO_2_)_24_O_4_(OH)_12_(CO_3_)_36_](HCO_3_)_4_(H_2_O)_88_, which contains ewingite-like uranyl carbonate nanoclusters, thus linking preparative chemistry with descriptive mineralogy.

## 2. Materials and Methods

### 2.1. Synthesis

Caution: *While isotopically depleted U was used in these experiments, precautions for handling radioactive materials should be followed*.

An aqueous solution of 0.0725 g (0.14 mmol) of uranyl nitrate dissolved in 2 mL of deionized distilled water was heated on a hot plate at 70 °C for 4–5 min. Then, 0.074 g (0.7 mmol) of CaCl_2_ and 0.095 g (1.0 mmol) of (NH_4_)_2_CO_3_ were added to the hot solution, which was stirred until all solid dissolved. The resulting yellowish transparent solution, which yielded a pH of 7, was left to evaporate in a watch glass at room temperature. Crystals of the synthetic ewingite-like compound (**SE**) begin to form after 12–18 h as the first precipitate, mostly in the form of fine- or cryptocrystalline powders with rare individual dendritic aggregates (Figure 1a,b). Shortly thereafter, massive crystallization of calcite started, making it almost impossible to select pure **SE**. This process was slowed by reducing the temperature to 4–5 °C, which also yielded slightly larger dendritic aggregates with single crystalline dendrite tips separated by up to 10–20 μm (single findings up to 50 μm). The primary and relatively pure crystallization of **SE** was observed in a laboratory without intensive ventilation without intensive ventilation, without air conditioning system and without heaters, respectively.

### 2.2. Chemical Composition

The chemical analysis of **SE** was carried out with a Hitachi FlexSEM 1000 (Tokyo, Japan) scanning electron microscope (Figure 1c) equipped with an EDS Xplore Contact 30 detector and Oxford AZtecLive STD (Oxford, UK) system of analysis. The accelerating voltage was 20 kV and beam current was 5 nA. Only Ca, U, C, and O were detected, with elements with atomic numbers higher than that of beryllium below detection limits. The following standards and X-ray lines were used: Ca—CaF_2_, K_α_; U—UO_2_, M_β_.

The chemical composition of **SE** is (wt.%, CO_2_ and H_2_O calculated based on structure): CaO 9.98, UO_3_ 58.90, CO_2_ 15.06, H_2_O 16.19, total 100.13. The empirical formula based on 278 O *apfu* is Ca_20.79_U_24.07_C_40_H_210_O_278_, or, taking into consideration the structural data, Ca_20.79_(H_3_O)_6_U_24.07_O_52_(CO_3_)_36_(HCO_3_)_4_(OH)_12_(H_2_O)_88_ (Figure 1c). N, Na, and K were not detected, resulting in the assignment of hydronium cations H_3_O^+^ as counter-ions in the structure of **SE**.

### 2.3. X-ray Diffraction Studies

The crystal structure of **SE** was determined from a greenish-yellow translucent dendrite tip with the dimensions 52 × 23 × 21 μm^3^ using single-crystal X-ray diffraction data collected by a Bruker Kappa Apex II Duo diffractometer at room temperature (microfocused MoKα radiation; 50 kV/0.6 mA; frame width 0.5°; exposure time 360 s/frame; Madison, WI, USA). Diffraction data were processed in the *CrysAlisPro* [14] program. The crystal structure of **SE** was solved by the dual-space algorithm and refined using the *SHELX* [15,16] programs incorporated in the *OLEX2* [17] program package. The crystal structure of **SE** is tetragonal (*P*-4*n*2; *a* = 24.6098(2), *c* = 24.6246(4) Å; *V* = 14,913.7(3) Å^3^; *Z* = 2; *R*_1_ = 0.060; CSD 2166564 [for cif see Appendix A]). The final model included coordinates and anisotropic displacement parameters for all non-H atoms. H atoms were not localized. The crystal structure of **SE** was refined as a 2-component twin using [010/-100/001] matrix with a statistically equal contribution of components (0.514(14)/0.486(14)).

Fragments of **SE** manually selected from synthesis products were used for the collection of powder diffraction data with a Rigaku Ultima IV diffractometer (CoKα radiation; 40 kV/30 mA; θ–θ Bragg–Brentano geometry; PSD D-Tex Ultra detector; Tokyo, Japan). A Rigaku SHT-1500 chamber was employed for experiments with **SE** in air over the temperature range of 25–800 °C; a Pt strip (20 × 12 × 2 mm^3^) was used as the heating element and sample holder. The temperature steps varied from 5 to 20 °C depending on the temperature range. The heating rate was 2 °C/min. The collection time at each temperature step was about 30 min. The irreversibility of the observed phase transformations was verified by collecting PXRD data following cooling.

Several fragments of **SE** were ground in an agate mortar for PXRD data collection using a Rigaku Ultima IV powder X-ray diffractometer (CuKα radiation; 40 kV/30 mA; θ–θ Bragg–Brentano geometry; PSD D-Tex Ultra detector; Tokyo, Japan) equipped with a background-free Si-single crystal sample holder at room temperature.

### 2.4. Infrared Spectroscopy

The IR spectrum of **SE** was recorded on a Bruker Vertex 70 (Bremen, Germany) spectrometer via the Attenuated Total Reflection Method in the region 4000–500 cm^−1^ (Figure 2). A MIRacle ATR accessory (Pike Technology; Fitchburg, WI, USA) with a Ge ATR crystal was used. IR spectroscopy is consistent with the structural data.

Infrared bands in the region above 2500 cm^−1^ (at 3590, 3420, 3230, 3130, 3000, 2950, 2900, 2890, 2830, 2730, and 2580 cm^−1^) are attributed to the ν O–H stretching frequencies of structurally nonequivalent H_2_O molecules forming the H-bonding network in the structure of SE. According to Libowitzky [18], the set of bands in this region corresponds to the approximate O–H∙∙∙O hydrogen bond lengths ranging between 2.6 and 3.2 Å. A band observed at 1645 cm^−1^ is attributed to the ν_2_ (δ) bending vibrations of symmetrically distinct H_2_O molecules. Bands in the region from 1570 to 1200 cm^−1^ (1570, 1525, 1487, 1445 and 1402 cm^−1^) are assigned to ν_3_ antisymmetric stretching vibrations of structurally nonequivalent CO_3_ planar groups. The splitting of the ν_3_ bands indicates the bidentate bonding of carbonate groups to uranyl ions in the crystal structure. The weak band at 1065 cm^−1^ is assigned as ν_1_ symmetric stretching vibrations of CO_3_ groups. The band caused by ν_2_ (δ) out-of-plane bending vibrations of CO_3_ groups is at 833 cm^−1^. Note that the ν_2_ (δ) CO_3_ bending vibrations may overlap with the ν_1_ UO_2_^2+^ symmetric stretching vibrations. Bands at 739, 711, 687, 671, and 665 are assigned to the ν_4_ (δ) in-plane bending CO_3_ vibrations.

Two strong bands at 891 and 873 cm^−1^ are attributed to the ν_3_ antisymmetric stretching modes of UO_2_^2+^.

## 3. Results and Discussion

### 3.1. Structure Descriptions

There are six crystallographically independent U^6+^ cations in the structure of **SE**, each of which is strongly bonded to two O^2−^ atoms, forming approximately linear O^2−^≡U^6+^≡O^2−^ uranyl cations (*Ur*) with U^6+^≡O^2−^ bond lengths ranging from 1.69(2) to 1.801(16) Å. The *Ur*1, *Ur*2, and *Ur*3 ions are equatorially coordinated by five oxygen atoms each, resulting in the formation of pentagonal bipyramids (U1–3–O_eq_ = 2.232(16)–2.443(17) Å), whereas the *Ur*4, *Ur*5, and *Ur*6 ions are coordinated by six oxygen atoms belonging to three carbonate groups, forming hexagonal bipyramids (U4–6–O_eq_ = 2.40(2)–2.458(18) Å). There are ten unique carbonate groups in the structure of **SE**, nine of which link to the equatorial edges of U-centered hexagonal bipyramids (C–O = 1.19(3)–1.35(4) Å), while the tenth group is not linked to U atoms, and due to the presence of one elongated and two short C–O bonds (1.47(4), 1.27(4) and 1.29(4) Å) is defined as an (HCO_3_)^–^ group.

There are two types of fundamental building units (FBU) in the structure of **SE**. FBU-1 consists of three *Ur*1–3 pentagonal bipyramids sharing common equatorial edges, with one μ^3^-O atom common to all three bipyramids, forming a trimeric unit (Figure 3a). The O atoms shared between two adjacent uranyl ions are hydroxyl groups, while the O atoms of the equatorial planes that do not bridge uranyl ions are from monodentate carbonate groups that link the FBU-1 and FBU-2 units. The FBU-2 unit is formed by the *Ur*4, *Ur*5, and *Ur*6 hexagonal bipyramids, in which each uranyl ion shares edges with three (CO_3_)^2−^ triangular groups (Figure 3b) to form uranyl tricarbonate clusters (UTC) that are common U-bearing structural units in uranyl carbonates [19]. Four FBU-1 and twelve FBU-2 are arranged to form a complex uranyl carbonate [(UO_2_)_24_O_4_(OH)_12_(CO_3_)_36_]^44–^ supertetrahedral nanocluster (Figure 3c,d). The cluster can be described as a combination of a tetrahedron and a cube, where the FBU-1 trimers correspond to faces of the tetrahedron, whereas two FBU-2 units delineate elongated faces of the cube (Figure 3e).

Isolated uranyl carbonate supertetrahedral nanoclusters in **SE** are separated by a network of Ca^2+^ cations and H_2_O molecules, both of which also occur inside the cluster. Five of the eight Ca sites are fully occupied. The Ca–O bond lengths range from 2.09(3) to 2.717(17) Å. Disorder (site splitting) and partial occupancies occur for the H_2_O sites. Calcium cations do not directly bridge between clusters. A complex H-bonding network involving interstitial H_2_O molecules, H_3_O^+^ ions, and (HCO_3_)^–^ groups link supertetrahedral nanoclusters into the crystal structure of **SE**.

There are several notable differences between the crystal structures of ewingite and **SE**. Ewingite contains both Ca and Mg, the latter of which does not occur in **SE**. Although tetragonal, the unit cell of ewingite is larger than SE (*I*4_1_/*acd*; *a* = 35.142(2), *c* = 47.974(3) Å; *V* = 59,245(8) Å^3^) [13]. However, the arrangement of supertetrahedral nanoclusters in both structures is nearly identical and can be described as a body-centered cubic (α-Fe type) packing with the distances between the centers of clusters at ~21.3 Å (Figure 4). The structure symmetries are related through the following sub-/supergroup sequence: *P*-4*n*2 → *I*-42*d* (a*’* = 2a, b*’* = 2b, c*’* = 2c) ← *I*4_1_/*acd*—which explains the eightfold difference in the unit-cell volume of ewingite compared to **SE**. Symmetry breaking in ewingite may be attributed to the distribution of Ca and Mg ions, as well as to the higher H_2_O content.

The most significant difference between the two structures is that supertetrahedral nanoclusters in the structure of ewingite are formed from three FBUs: four trimers of pentagonal bipyramids, six uranyl tricarbonate, and six uranyl bicarbonate moieties [13]. The structure of **SE** has 12 uranyl tricarbonate units. It cannot be excluded that, due to the low quality of the natural ewingite crystals studied previously, some atoms of carbonate groups were not located in the difference-Fourier maps.

The similarity of the structural architectures of both compounds is clearly seen from the comparison of their powder XRD patterns (Figure 5a,b), where the overlapping basal peaks correspond to the supertetrahedral nanocluster packing. It should be also noted that slight grinding in an agate mortar results in the reduction in the unit cell observed in the XRD pattern from the shift of the basal reflections, which are responsible for the general cluster arrangement, to the far angular region, while reflections responsible for more subtle structural interactions are extinct (Figure 5c).

### 3.2. Structural Complexity Analysis

Ewingite is the mineral with the most complex crystal structure known to date and possesses the highest amount of structural information observed in minerals. The theory of structural complexity was developed by S.V. Krivovichev [20,21,22,23] and successfully implemented in recent works [24,25,26,27]. Olds et al. proposed that the information content of ewingite is *c.a*. 23,000 bits/cell [13]. Our recent consideration of the disorder of sites and the H atom assignment indicate that the structural complexity of ewingite is likely in the range of 19,500–21,500 bits/cell, depending on the hydration state [19]. The information-based complexity parameters for ewingite and **SE** are given in Table 1. Although the hydration state of **SE** probably may also vary, the structural complexity of the synthetic compound is half that of ewingite at 9515.770 bits/cell. The contribution of factors such as the H-bonding system, interstitial substructure, and cluster stacking to the overall structural complexity [28] in both compounds is comparable, which points to the similarity of substructural units’ arrangement and their role in ewingite and **SE**. The most significant difference is the topological complexity of the cluster, which is attributed to the aforementioned absence of specific carbonate groups in the structural model of ewingite. The tendency for the complexity parameters of synthetic compounds to usually be lower than those of minerals with identical or genetically similar topologies of U-bearing complexes was recently described for uranyl sulfates [29,30] and uranyl selenites [31] and seems to be quite common in general. On the other hand, the complexity of a crystal structure contributes negatively to the configurational entropy of the crystalline phase [32], although metastable polymorphs are usually structurally simpler than their more thermodynamically stable counterparts [33,34].

According to visual observation and experimental studies, in terms of their preservation, ewingite is more stable in air than **SE**, in which polycrystalline aggregates undergo degradation even with insignificant mechanical stress, which can be seen from the experimental PXRD pattern (Figure 5c). The indexation of the **SE** PXRD pattern by the Pawley method using the *TOPAS* software [35] resulted in the following unit cell: tetragonal; *a* = 23.12(1), *c* = 22.92(3) Å (Figure 6a). The high-temperature PXRD experiment demonstrates that **SE** readily loses its crystallinity, and traces from the basal reflections, which correspond to the uranyl carbonate nanocluster packing, disappear entirely after 80 °C (Figure 6b). Later, starting at *c.a*. 420 °C under continuous heating, the CaUO_4_ (PDF 01-085-0577; ICDD PDF-2 Database, release 2019 [36]) phase crystallizes.

The recent findings relating to the unnamed phase, equivalent to a now-described synthetic compound, at several localities in nature (Jáchymov, Czech Republic; Annaberg, Germany; Red Canyon district, Colorado, USA) have documented that the **SE** occurred there without the presence of ewingite (unpublished observations of J.P.). Furthermore, the first chemical analyses of the natural material also suggested a large chemical variability (namely in the Ca:Mg ratio), while the single-crystal X-ray diffraction patterns are very similar and also similar to those of **SE**. Nevertheless, these findings require deeper study, which is beyond the scope of this paper.

Keeping in mind the similar architecture of ewingite and **SE** as well as findings relating to the novel natural phase, it can be assumed that the formation of a particular crystalline form is highly dependent on the chemical composition of ground waters. On the other hand, since ewingite is apparently more stable, it is plausible that the **SE** phase can be regarded as a primary and/or metastable reaction product which further re-crystallizes into a more stable form under environmental conditions.

## Figures and Tables

**Figure 1 materials-15-06643-f001:**
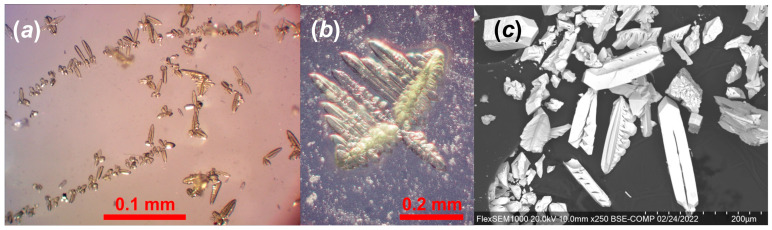
The onset of crystallization of the synthetic ewingite-like compound (**SE**) (**a**), a dendritic aggregate of **SE** surrounded by fine crystalline calcite powder (**b**), and an SEM image of **SE** crystals (**c**).

**Figure 2 materials-15-06643-f002:**
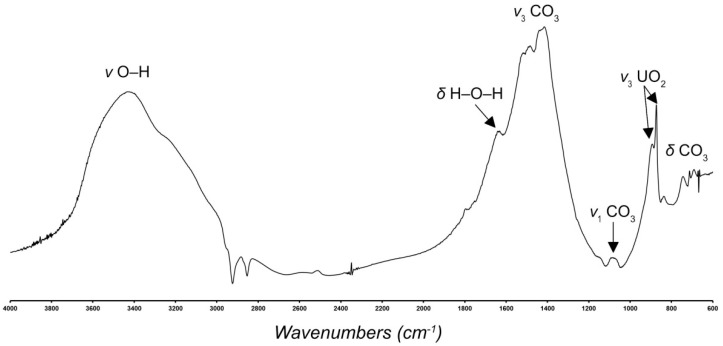
Powder infrared absorption spectrum of the **SE**.

**Figure 3 materials-15-06643-f003:**
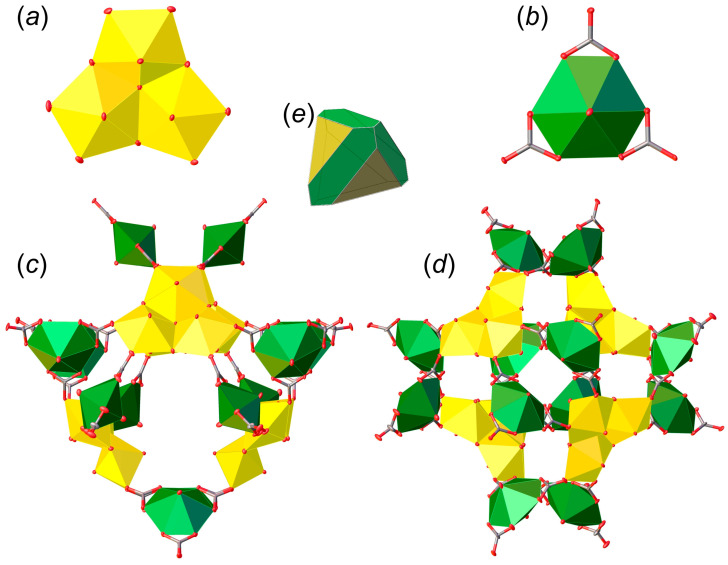
Fundamental building units (FBU) in the structure of **SE** (**a**,**b**), and their arrangement into a supertetrahedral nanocluster (**c**,**d**) that is similar to the combination of a tetrahedron and a cube (**e**). Legend: uranyl pentagonal bipyramids = yellow; uranyl hexagonal bipyramids = green; different coloring of *Ur* polyhedra is used for clarity; red = O atoms; grey = C atoms.

**Figure 4 materials-15-06643-f004:**
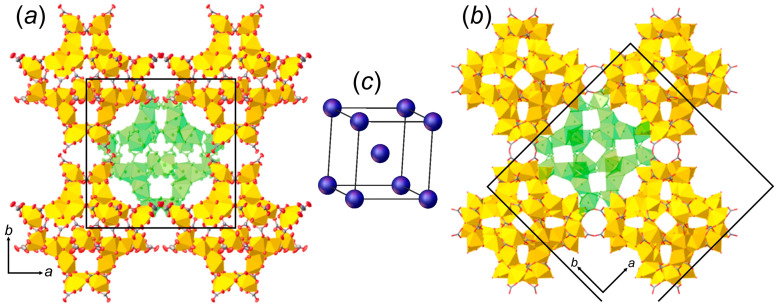
Arrangement of supertetrahedral nanoclusters in the structures of **SE** (**a**) and ewingite (**b**), which is similar to the body-centered cubic (α-Fe type) packing (**c**). Interstitial ions and molecules are omitted for clarity; central nanocluster in the body-centered cubic packing is green-colored. Legend: U-centered polyhedra = yellow; red = O atoms; grey = C atoms.

**Figure 5 materials-15-06643-f005:**
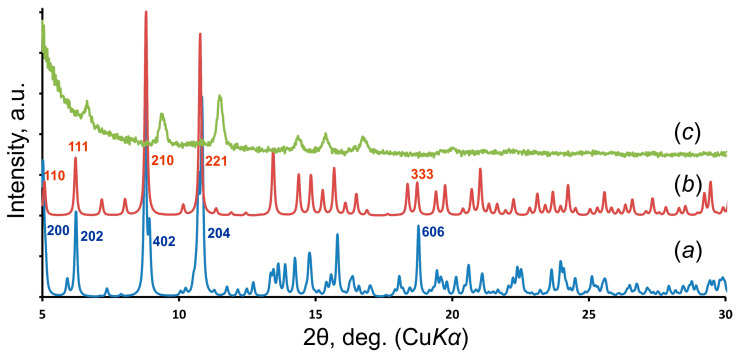
PXRD patterns calculated from the structural data of the ewingite (**a**), **SE** (**b**), and pattern collected during the PXRD measurement of ground **SE** sample (**c**).

**Figure 6 materials-15-06643-f006:**
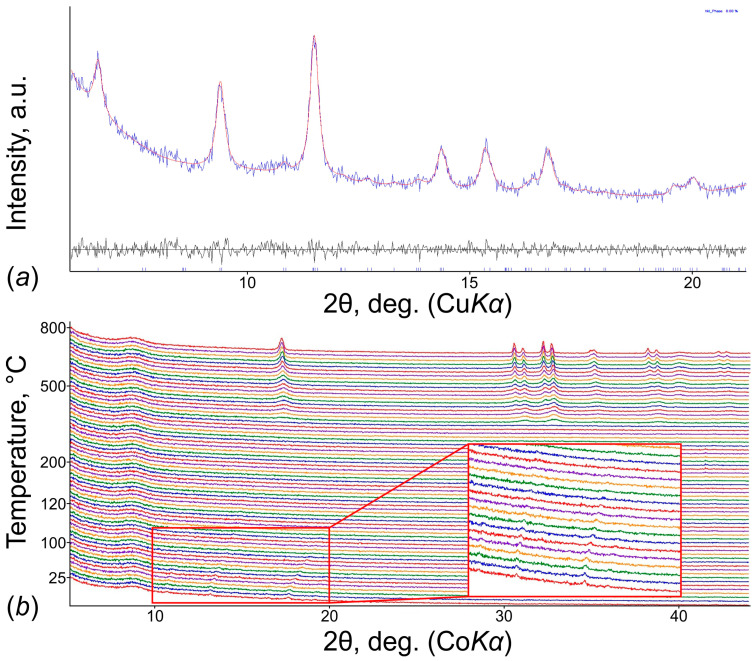
PXRD patterns of **SE** as a function of temperature (25–800 °C) under heating in air (**a**) and indexation of the **SE** PXRD pattern (**b**).

**Table 1 materials-15-06643-t001:** Structural complexity parameters for ewingite and **SE**.

Complexes That Contribute to Structural Complexity [27]	Ewingite [18]			SE		
	*υ*, Atoms	*I_G,total_*, Bits/Cell (*I_G_*, Bits/Atom)	Contribution, %	*υ*	*I_G,total_*, Bits/Cell (*I_G_*, Bits/Atom)	Contribution, %
Topological complexity of the cluster	220	1271.820 (5.781)	6.9	244	1447.100 (5.931)	15.2
Structural complexity of the cluster	220	1271.820	0	244	1447.100	0
Stacking of clusters	880	3813.886	20.8	488	1506.180	15.8
Interstitial structure	588	4531.056	24.7	318	2499.190	26.3
H-bonding	1040	8717.650	47.6	484	4063.300	42.7
Structural complexity of the entire structure	2508	18,335.988 (7.311)	100	1290	9515.77 (7.377)	100

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
