# Peer review of "One of Nature’s Puzzles Is Assembled: Analog of the Earth’s Most Complex Mineral, Ewingite, Synthesized in a Laboratory"

_materials, 2022, doi:10.3390/ma15196643_

Round 1

Reviewer 1 Report

This paper deals with the synthesis, crystallisation and crystals structure study of an analog of the complex mineral Ewingite. Even though the paper could be published as is, since it is well-written and the study well documented and described, several points could make this paper better.

This analog of Ewingite crystalizes solely with Ca cations, due to the synthesis parameters omitting Mg, even though the natural mineral is a Mg/Ca combination. It would be interesting for the reader to know if the choice of omitting Mg was intentional or if synthesis attempts have been done, but to no avail concerning a crystalline product. A study of the complete Mg/Ca family could give an interesting result concerning the changes of symmetry in the crystal structure with the different ratios, i.e: would the symmetry be different with a pure Mg compound ?

It would be interesting for the study to follow the pH of the solution while the crystallisation is happening, in order to try and inhibit the growth of calcite that prevent large single crystals to be grown. What are the parameters of the mother liquor during crystal growth ?

The crystal structure resolution is a challenge with the size of the unit cell and the sheer number of atoms, however, it would seem that one might get better results at lower temperatures (cooling with nitrogen systems for example). Since the TWIN command has been used to solve this crystals structure, a calculated (hkl) of choice would be interesting to see, as well as the % of the different twins (in the text, not only in the cif file).

Concerning the PXRD, the temperature study does not bring much to the paper, but a study at different stages of grinding might help in understanding the stability of the structure. Perhaps a calculation of the powder parameters and/or structure refinement at different grinding stages could prove insightful ?

Overall, I recommend this paper for publication, but more in-depth studies of certain aspects of this analog would make this paper better.

Author Response

First of all, we want to thank the Reviewer for the careful reading of our manuscript and for its high evaluation.

Q1: This analog of Ewingite crystalizes solely with Ca cations, due to the synthesis parameters omitting Mg, even though the natural mineral is a Mg/Ca combination. It would be interesting for the reader to know if the choice of omitting Mg was intentional or if synthesis attempts have been done, but to no avail concerning a crystalline product. A study of the complete Mg/Ca family could give an interesting result concerning the changes of symmetry in the crystal structure with the different ratios, i.e: would the symmetry be different with a pure Mg compound?

A1: Thanks for this comment. Indeed, our synthetic experiment contains only Ca cations. We’ve tried to conduct other experiments on Mg/Ca and pure Mg systems as well, but didn’t get any large enough single crystals. And this is (was) the main our target. Of course, we continue working in this field and hoping that we will find pathways to obtain good crystalline material.

Q2: It would be interesting for the study to follow the pH of the solution while the crystallisation is happening, in order to try and inhibit the growth of calcite that prevent large single crystals to be grown. What are the parameters of the mother liquor during crystal growth?

A2: The resulted solution yielded a pH of 7. Information was added to the text. Techniques that were yet tried partially helped to inhibit calcite growth, but also resulted in the absence of visible SE crystals formation (only cryptocrystalline or amorphous powders).

Q3: The crystal structure resolution is a challenge with the size of the unit cell and the sheer number of atoms, however, it would seem that one might get better results at lower temperatures (cooling with nitrogen systems for example). Since the TWIN command has been used to solve this crystals structure, a calculated (hkl) of choice would be interesting to see, as well as the % of the different twins (in the text, not only in the cif file).

A3: First XRD experiments were made at 100 K nitrogen flow, however the diffraction data were very poor. Couple experiments at room temperature gave us much better data (including current one). May be it was just a coincidence and quality of those crystals were poor regardless of cooling, but since there are not too much crystals that can be used for SC XRD data collection, up to now we decided to use the data collected at room temperature. Information on twin refinement was added to the text.

A4: Concerning the PXRD, the temperature study does not bring much to the paper, but a study at different stages of grinding might help in understanding the stability of the structure. Perhaps a calculation of the powder parameters and/or structure refinement at different grinding stages could prove insightful?

Q4: Indeed. We hoped that high temperature PXRD scans will give us more data, but even though it shows us stability of general cluster arrangement. Thanks for the idea on the correlation with grinding. We’ll try to implement it.

Reviewer 2 Report

In this paper, the authors report the synthesis and characterization of the novel synthetic compound that contains ewingite-like uranyl carbonate nanoclusters. Overall, this paper is well-written and well-organized, and I recommend its acceptance to Materials. Before publication, the following minor points should be revised:

1. The authors mentioned that slight grinding in an agate mortar results in the reduction of the unit cell observed in the XRD pattern from the shift of the basal reflections to the far angular region. See Figure 5c, in addition to obvious shifting, there are still have some peaks is missing, maybe some discussions should be given.

2. It is better to give more discussion about the determination of crystal structure of the title compound, now in the current version, necessary detail is still no found.

3. The color for each atoms should be given for Figures 3 and 4.

Author Response

We want to thank the Reviewer for the careful reading of our manuscript and for its high evaluation.

In this paper, the authors report the synthesis and characterization of the novel synthetic compound that contains ewingite-like uranyl carbonate nanoclusters. Overall, this paper is well-written and well-organized, and I recommend its acceptance to Materials. Before publication, the following minor points should be revised:

Q1: The authors mentioned that slight grinding in an agate mortar results in the reduction of the unit cell observed in the XRD pattern from the shift of the basal reflections to the far angular region. See Figure 5c, in addition to obvious shifting, there are still have some peaks is missing, maybe some discussions should be given.

A1: We believe that only “basal” i.e. those reflections that are responsible for the tetrahedral clusters general arrangement remain on the pattern. While others, as you rightly pointed out, disappear. This information can be found in the text.

Q2: It is better to give more discussion about the determination of crystal structure of the title compound, now in the current version, necessary detail is still no found.

A2: Additional information on structure refinement was added to the text.

Q3: The color for each atoms should be given for Figures 3 and 4.

A3: Done.

Reviewer 3 Report

see attached file

Author Response

We want to thank the Reviewer for the careful reading of our manuscript and for its high evaluation.

Q1: The authors report the synthesis and the structural characterization of a novel uranyl carbonate and consider analog to the mineral ewingite. As admitted by the authors in the section Structure description, actually the analogy between the synthetic compound and ewingite is more apparent (Fig. 4) than real. In fact, the two compounds significantly differ both in the chemical composition and in the structural connection of a different number of structural building units. Taking also into account that the published crystal structure of ewingite is very poorly determined, it would be sensible to avoid the present emphasis on rather approximate analogies between the two compounds; in particular, the title must be less emphatic. A title like the following one is suggested: Synthesis and crystal structure of a novel uranyl carbonate. Comparison with the most complex mineral ewingite. Consequently, the text should be revised.

A1: Of course, partially you’re right. But we can’t totally agree with your comments. First of all, we believe, the main achievement of this work is that we could obtain in laboratory conditions these uranyl-carbonate nanoclusters, which were only previously found in the crystal structure of the only natural phase. And it exactly means that we were able to get as close to the natural synthesis pathway as possible. Yes, the chemistry differs, but not so crucially, since the nano-cluster arrangement is absolutely similar. This work should be regarded as the first of the series (we hope we would be able to expand the knowledge about these compounds), but the fact that we’ve got such compound albeit a little, but still allow us to give such a loud title. We really hope that you will forgive us this slight exaggeration and wouldn’t strictly insist on changing the title.

Q2: Line 35 – A reference to De re metallica by Agricola would be useful.

A2: Thanks for this suggestion. Done.